# Urodynamics Are Essential to Predict the Risk for Upper Urinary Tract Damage after Acute Spinal Cord Injury

**DOI:** 10.3390/biomedicines11061748

**Published:** 2023-06-17

**Authors:** Veronika Birkhäuser, Collene E. Anderson, Marko Kozomara, Mirjam Bywater, Oliver Gross, Stephan Kiss, Stephanie C. Knüpfer, Miriam Koschorke, Lorenz Leitner, Ulrich Mehnert, Helen Sadri, Ulla Sammer, Lara Stächele, Jure Tornic, Martina D. Liechti, Martin W. G. Brinkhof, Thomas M. Kessler

**Affiliations:** 1Department of Neuro-Urology, Balgrist University Hospital, University of Zürich, 8008 Zürich, Switzerland; 2Swiss Paraplegic Research, 6207 Nottwil, Switzerland; 3Department of Health Sciences and Medicine, University of Lucerne, 6002 Lucerne, Switzerland; 4Department of Urology, Cantonal Hospital Lucerne, 6000 Lucerne, Switzerland; 5Department of Urology, Cantonal Hospital Aarau, 5001 Aarau, Switzerland; 6Department of Neuro-Urology, Clinic for Urology, University Hospital Bonn, 53127 Bonn, Germany; 7Department of Urology, Winterthur Cantonal Hospital, 8400 Winterthur, Switzerland

**Keywords:** neuro-urology, neurogenic lower urinary tract dysfunction, urodynamics, spinal cord injuries, urinary bladder, neurogenic, urinary bladder, overactive, decision support techniques, clinical decision-making, predictive value of tests

## Abstract

We used clinical parameters to develop a prediction model for the occurrence of urodynamic risk factors for upper urinary tract (UUT) damage during the first year after acute spinal cord injury (SCI). A total of 97 patients underwent urodynamic investigation at 1, 3, 6, and 12 months after acute SCI, within the framework of a population-based longitudinal study at a single university SCI center. Candidate predictors included demographic characteristics and neurological and functional statuses 1 month after SCI. Outcomes included urodynamic risk factors for UUT damage: detrusor overactivity combined with detrusor sphincter dyssynergia, maximum storage detrusor pressure (pDetmax) ≥ 40 cmH_2_O, bladder compliance < 20 mL/cmH_2_O, and vesicoureteral reflux. Multivariable logistic regression was used for the prediction model development and internal validation, using the area under the receiver operating curve (aROC) to assess model discrimination. Two models showed fair discrimination for pDetmax ≥ 40 cmH_2_O: (i) upper extremity motor score and sex, aROC 0.79 (95% CI: 0.69–0.89), C-statistic 0.78 (95% CI: 0.69–0.87), and (ii) neurological level, American Spinal Injury Association Impairment Scale grade, and sex, aROC 0.78 (95% CI: 0.68–0.89), C-statistic 0.76 (95% CI: 0.68–0.85). We identified two models that provided fair predictive values for urodynamic risk factors of UUT damage during the first year after SCI. Pending external validation, these models may be useful for clinical trial planning, although less so for individual-level patient management. Therefore, urodynamics remains essential for reliably identifying patients at risk of UUT damage.

## 1. Introduction

One of the most frequent sequelae of spinal cord injury (SCI) is neurogenic lower urinary tract dysfunction (NLUTD), which represents a considerable social and economic burden [1,2,3]. NLUTD is associated with urological morbidity, such as urosepsis, urinary incontinence, and the risk of deteriorating renal function and has a significant negative impact on health-related quality of life [1,2]. Consequently, recovery of bladder function is often identified as one of the top priorities in patients with SCI [4]. Due to the potential for changes in neurological and lower urinary tract (LUT) function during the first year after SCI, neuro-urological management aims to maintain or improve upper urinary tract (UUT) and LUT function and to identify and treat patients with risk factors for urological morbidity [5]. In order to achieve these aims, emphasis is placed on ensuring efficient bladder emptying and maintaining safe storage pressures [5]. There is an established body of evidence indicating that high maximum detrusor pressure (pDetmax), ≥40 cmH_2_O, during the storage phase is a key risk factor for UUT complications and renal failure [6,7,8,9]. The gold standard to assess LUT symptoms, evaluate and monitor LUT function and storage pressures, identify patients with a risk for UUT deterioration, and also to guide and monitor treatment, is video-urodynamic investigation [2,10]. However, urodynamic investigation (UDI) is perceived to have some disadvantages: it is relatively invasive, it requires specialized equipment and expertise, and it is associated with high financial costs [11].

Prognostic models for urological outcomes could support clinical decision-making, thereby promoting stratified management [12], and potentially reducing the dependence on UDIs. The prediction of future outcomes provided by prognostic models also can support patient counseling and planning, as well as allow for patient stratification in randomized controlled trials [12]. A simple model using lower extremity motor score (LEMS), a clinical parameter derived from International Standards for Neurological Classification of Spinal Cord Injury (ISNCSCI) assessments, was proposed by the European Multicenter Study on Spinal Cord Injury (EMSCI) study group to predict urinary continence and complete bladder emptying 1 year after a traumatic SCI [13]. This predictive model was validated using external data from the National SCI Database (NSCID, Birmingham, AL USA) [14]. Currently, there are no prognostic models to identify patients with an acute SCI who are at elevated risk of upper urinary tract damage. As early identification of such patients is essential for proactive and individualized urological management within the first year after SCI, we aimed to develop a clinical prediction model for established urodynamic risk factors for urological morbidity, i.e., detrusor overactivity combined with detrusor sphincter dyssynergia (DO–DSD), pDetmax ≥ 40 cmH_2_O during the storage phase, bladder compliance < 20 mL/cmH_2_O, and vesicoureteral reflux (VUR).

## 2. Materials and Methods

### 2.1. Patients

This patient cohort and its urological evaluation and management have previously been described in detail [15,16]. The population is composed of patients that participated in the prospective, longitudinal, and population-based EMSCI study (www.emsci.org (accessed on 23 May 2023)) between January 2014 and December 2019 and underwent video-UDI at a SCI-specialized university hospital in Switzerland. Inclusion criteria were patients aged ≥ 18, with an acute SCI from a single traumatic or ischemic event, who had their first neurological assessment within 40 days after a SCI. Exclusion criteria were a severe craniocerebral injury or cognitive impairment, polyneuropathy, or pre-existing dementia. The participants underwent assessments at 1 month (days 16–40), 3 months (days 70–98), 6 months (days 150–186), and 12 months (days 300–546) after SCI. Written informed consent was obtained from all participants and the local ethics committee (PB_2016-00293, EK-03/2004) approved the study.

### 2.2. Neuro-Urological Evaluation and Management

Neuro-urological assessments were performed as described previously [1]. Video-UDI was conducted according to the International Continence Society (ICS) recommendation on good urodynamic practice [17], preferably in a sitting position, using a multichannel urodynamic system, as same-session repeat filling cystometry and pressure flow study. ICS recommendations for definitions, methods, and units were applied [18]. In order to minimize the risk of assessor bias, all UDIs were randomly assigned to two expert neuro-urologists for evaluation on a per-patient basis [16]. Urological management was in line with the European Association of Urology (EAU) Guidelines on Neuro-Urology [2], and a table describing the bladder emptying method and use of medications with a potential effect on the bladder at each timepoint can be found in a previous open access publication from our group [15].

### 2.3. Predictive Measures

The Transparent Reporting of a multivariable prediction model for Individual Prognosis Or Diagnosis (TRIPOD) statement was used to guide model development and reporting [19]. Generally, prognostic modeling approaches focus on identifying the combination of independent factors that most accurately predicts which patients will experience the outcome of interest [12,19]. In this case, the outcomes of interest were the unfavorable urodynamic parameters: DO–DSD, pDetmax ≥ 40 cmH_2_O during the storage phase, bladder compliance < 20 mL/cmH_2_O, and VUR. As this dataset contains a relatively small number of subjects, an evidence-based strategy was used to select a limited number of candidate predictors, as opposed to relying on data-driven procedures for model selection. Values for predictors were taken from the one-month post-SCI timepoint to allow for early identification of high-risk patients. Candidate predictors included demographic characteristics (age and sex), and commonly used measures for neurological status, and functional independence. In particular, predictors from the EMSCI models [13] were examined (in the results reported as models 1 and 2), as it would be ideal to have a common prognostic model that provides an excellent prediction of an elevated risk of upper urinary tract damage, in addition to urinary continence and complete bladder emptying. Furthermore, predictors previously identified as potentially relevant using this dataset were included in the current analysis [16]. Predictors that might be subject to external influences one month after SCI, such as acute care referral patterns, were not included as candidates (e.g., bladder emptying method). Neurological and functional evaluation used the ISNCSCI [20] and the Spinal Cord Independence Measure Version III (SCIM III) [21] assessments. The ISNCSCI provided data on a neurological level, SCI severity (American Spinal Injury Association Impairment Scale (AIS) grade), upper extremity motor score (UEMS), LEMS, and light-touch scores [20]. The SCIM assesses functional independence in patients with SCI and is divided into three subdomains: self-care, respiration, and sphincter management and mobility [21].

### 2.4. Statistical Analyses

Univariable testing included Chi-squared and Fisher’s exact tests for categorical variables and nonparametric tests (Kruskal–Wallis) for continuous variables, as all had non-normal distributions. Logistic regression models were used to evaluate candidate predictors of storage, pDetmax ≥ 40 cmH_2_O. Further analysis based on logistic regression modeling was not applied to DO–DSD, VUR, or any unfavorable urodynamic parameter due to the presence of complete or quasi-complete separation (i.e., less than two patients were ascribed to a given outcome category for any categorical variable including sex, AIS grade, and neurological level). Model discrimination was evaluated using the area under the receiver operating characteristic curve (aROC). AIS grades B and C were combined to provide stable estimates due to the small number of subjects in these categories; this category combination was based on the cross-tabulation of AIS grade against the outcome of interest, with post hoc likelihood ratio tests used for confirmation. For continuous variables (age, UEMS, LEMS, SCIM respiration, and sphincter management subscale), the statistical support for a nonlinear relationship was also evaluated using multivariable fractional polynomials [22]. For model selection, candidate predictors were examined individually and in combination using logistic regression models with a visual inspection of the ROC curves, along with supplementary Lasso regression to confirm the variable selection and account for multicollinearity. Internal validation was performed using a bootstrapping approach (1000 iterations) [23], optimism-corrected C-statistics were used to evaluate model discrimination, the scale Brier score was used to indicate overall model performance, and the expected-to-observed ratio was used to assess model calibration [24].

To evaluate the prognostic discrimination of the models (as opposed to cross-sectional prediction), additional regression analyses were performed, which only included patients who had not already reached the endpoint (pDetmax ≥ 40 cmH_2_O) at the 1-month follow-up. The primary analyses used results derived from models implementing multiple imputations (MIs) with chained equations to account for missing data [25,26]. Sensitivity analysis additionally accounted for loss-to-UDI-follow-up and included models with inverse probability weights (IPWs), as well as an “informed” imputation model based on further information from the clinical record—usually clinical visits without UDI (see Appendix A, and Appendix A for missing data methodology). A final sensitivity analysis was run to test the performance of the models using a different outcome—pDetmax ≥ 40 cmH_2_O or the use of antimuscarinics within the first year after SCI. This extreme scenario gives insight into the generalizability of these models to contexts where access to treatment is limited or not as timely. Statistical analyses were performed in Stata (versions 16.1 and 17.0, College Station, TX, USA).

## 3. Results

### 3.1. Study Population and Prevalence of Unfavorable Urodynamic Parameters

The study population consisted of 97 patients who underwent at least 1 UDI after an acute SCI, of whom 73 (75%) returned for 12-month assessments. Of the 24 patients who were lost-to-UDI follow-up before the 12-month UDI, 4 patients declined further care, 8 were transferred to another clinic, 3 died, and for 9, the reason was unclear. Patient characteristics are shown stratified, according to the 12-month follow-up status in Table 1. In the univariable analysis, none of the candidate predictors were associated with a missed 12-month follow-up assessment; however, a missed 12-month assessment was associated with DO–DSD, pDetmax ≥ 40 cmH_2_O, and the occurrence of any unfavorable urodynamic parameter. Table 2 presents the results for DO–DSD, storage pDetmax ≥ 40 cmH_2_O, VUR, and any unfavorable urodynamics parameter within the first year after a SCI, stratified, according to the predictor variables. As the prevalence of DO–DSD, and by extension any unfavorable UDI parameter, was very high, the prevalence of VUR was very low, and there were no cases of bladder compliance < 20 mL/cmH_2_O; thus, the remaining prognostic modeling analyses focused on the pDetmax ≥ 40 cmH_2_O outcome.

In total, 38/97 patients (39%) had a pDetmax ≥ 40 cmH_2_O within one year after SCI, and 55% of these patients (21/38) already had reached this endpoint at the 1-month UDI (n = 7 missed the 1-month UDI). A total of 61 patients were treated with antimuscarinics during the first year after SCI, 29 of whom did not develop pDetmax ≥ 40 cmH_2_O. None of the patients were using antimuscarinics, beta-3 adrenergic agonists, or onabotulinumtoxinA at the time of the 1-month UDI. An ‘informed’ outcome status was derived for 13/20 patients who were lost to follow-up without reaching the outcome for use in the sensitivity analysis (Appendix A). Most of the informed loss-to-follow-up cases (12/13) were clinically judged to have a low risk of developing pDetmax ≥ 40 cmH_2_O.

### 3.2. Model Performance and Internal Validation

Model 1 used LEMS as the sole predictor [13]. The multiple imputation model with all observations (n = 97) did not provide adequate discrimination, with an aROC value of 0.53 (95% CI: 0.40–0.66) (Figure 1, Appendix A), nor did the model excluding the 21 patients who had already reached a pDetmax ≥ 40 cmH_2_O at the one-month time-point (n = 76), aROC 0.57 (95% CI: 0.41–0.75). Model 2 (LEMS, highest light touch score in the S3 dermatome, and the SCIM respiratory–sphincter subscale) [13] also resulted in poor discrimination, aROC 0.65 (0.52–0.78) (Figure 1, Appendix A). Full logistic regression results (adjusted odds ratios) from models 1–4 are presented in Appendix A.

A model that included UEMS and sex (model 3) showed fair discrimination, aROC 0.79 (95% CI: 0.69–0.89); further, observations that already had a pDetmax ≥ 40 cmH_2_O at the first measurement were excluded 0.73 (95% CI: 0.60–0.86) (Figure 1, Appendix A). Internal validation resulted in an optimism–corrected C-statistic of 0.78 (95% CI: 0.69–0.87), the scaled Brier score was 22.3%, and the expected-to-observed ratio was 0.997. Similarly, a model that included neurological level, AIS grade one month after SCI, and sex (model 4), showed fair discrimination, aROC 0.78 (95% CI: 0.68–0.89), and remained in the fair range after excluding the observations, which had already reached the endpoint at baseline aROC 0.73 (0.60–0.87) (Figure 1, Appendix A). Internal validation resulted in a C-statistic of 0.76 (95% CI: 0.68–0.85), a scaled Brier score of 17.3%, and an expected-to-observed ratio of 0.995. The predicted probability of reaching the outcome according to models 3 and 4 is displayed in Figure 2, full logistic regression results are presented in the Appendix A.

The most commonly selected model using the data-driven approach, Lasso regression, included sex, UEMS, and AIS grade and showed good discrimination with an aROC of 0.80 (95% CI: 0.70–0.90). However, internal validation revealed that the optimism–corrected C-statistic was 0.78 (95% CI: 0.69–0.87), the scaled Brier score was 21.7%, and the expected-to-observed ratio was 0.997, indicating that this model also provides ‘fair’ performance in the current dataset.

### 3.3. Sensitivity Analyses

Sensitivity analyses using complete cases analysis, IPWs and ‘informed’ outcome assignments yielded similar results to the primary analyses in terms of model discrimination (Appendix A). Further sensitivity analyses investigating the discrimination of the four models for the outcome pDetmax ≥ 40 cmH_2_O or starting antimuscarinics within the first year after SCI resulted in very similar aROC values and confidence intervals (Appendix A: Appendix A (regression output) and Appendix A (aROC results)).

## 4. Discussion

### 4.1. Main Findings

We aimed to develop a clinical model to predict the occurrence of urodynamic risk factors for UUT damage within the first year after SCI. While LEMS is highly predictive of urinary incontinence and complete bladder emptying one year after SCI [13], it did not adequately predict the pDetmax ≥ 40 cmH_2_O outcome. We identified two potential prognostic models based on the noninvasive, inexpensive, clinical routine ISNCSCI assessment: (i) UEMS and sex; (ii) neurological level, AIS grade, and sex. However, both models showed only a ‘fair’ predictive performance.

### 4.2. Findings in the Context of Existing Evidence

Timely identification of urodynamic risk factors, especially high pDetmax (≥ 40 cmH_2_O) during the storage phase, which allows the consecutive initiation of antimuscarinic therapy as a first-line treatment, is a cornerstone of urological management after SCI [5]. A reliable prognostic model could play a critical role in targeting patients who are likely to need intervention [12]. However, to the best of our knowledge, no such models currently exist. The need for an early-phase prognostic model is highlighted by recent studies using the current patient cohort, whereby a study found that 39% of the patients exhibited pDetmax ≥ 40 cmH_2_O within the first year after a SCI [15]. High storage pDetmax (≥40 cmH_2_O) was first identified, during a UDI performed 1 or 3 months after SCI, in 82% of the affected patients [16]. Of note, in the first year after a SCI, none of the patients in this cohort developed a low-compliance bladder [15,16]. Whether this is due to the treatment based on urodynamic findings or the short follow-up period could be addressed in future research. Furthermore, research that focuses on preventing the development of NLUTD after SCI gained importance during the last decade. Pilot studies in patients with acute SCI have shown that early bilateral sacral neuromodulation and transcutaneous tibial nerve stimulation potentially prevent the development of neurogenic detrusor overactivity (DO) and urinary incontinence [27], and higher detrusor pressure [28], respectively. These pilot studies served as a foundation for larger clinical trials aiming to conclusively evaluate the effect of early sacral neuromodulation [29] and transcutaneous tibial nerve stimulation [30] on the prevention of neurogenic DO. While the EMSCI LEMS models were found to be highly predictive of urinary continence and complete bladder emptying at one year after an acute SCI [13,14], based on the current results, we cannot recommend their use for identifying patients at risk of high pDetmax during the storage phase.

### 4.3. Implications—Practice

Optimization of patient counseling and patient-tailored urological management could be facilitated by a prognostic model that identifies patients with risk factors for UUT damage. Currently, there are no guidelines for a standardized UDI follow-up schedule during the first year after a SCI, although a UDI is considered the gold standard to objectively assess LUT function and to identify urodynamic risk factors that are associated with long-term urological complications and morbidity and represent a risk of UUT deterioration [2,5,6,7,8,9]. In the present study, we utilized a standardized UDI follow-up schedule consisting of a UDI at four predefined timepoints after a SCI; however, the UDI availability was limited in some settings. Our results indicate that males, persons with lower UEMS, cervical SCI, and AIS grades A, B, and C SCIs might potentially need more frequent UDI follow-ups after a SCI since they are at higher risk of high storage pDetmax and/or need antimuscarinic treatment. Therefore, management of these high-risk patients in specialized centers is highly warranted. As the models reported here only provide fair prediction, they do not provide sufficient information for individual-level patient diagnosis. Even in the model for volitional voiding proposed by the EMSCI group with excellent prediction [13], there was a substantial proportion of patients (17%) with a LEMS of 50 who could not void volitionally, while 1.5% of those with a LEMS of 0 could void volitionally [14]. These findings highlight the importance of UDIs to objectively assess LUT function, and consequently, counsel patients.

### 4.4. Implications—Research

One important application of prognostic models is for risk stratification in clinical trials, especially in fields such as SCI, with limited patient numbers, whereby such models can improve the study design and analysis of clinical trials [12]. An especially important direction for future research would be an investigation into the predictive value of the 1-month UDI.

### 4.5. Strengths and Limitations

To the best of our knowledge, this is the first attempt to produce a prognostic model for the occurrence of urodynamic risk factors of UUT damage after a SCI. In the current study, the urodynamic investigations took place irrespective of the spinal shock status. This is in contrast to some guidelines that recommend performing UDIs after the resolution of spinal shock [10,31] since the detrusor might present with hypocontractility or acontractility during the spinal shock phase. However, early investigations can be beneficial since the duration of the spinal shock period is not well defined [5], and during the early phase after SCI, a high percentage of patients already show unfavorable urodynamic parameters [15,32]. Although the standardized UDI schedule identified more patients with the outcomes during the first year after SCI compared to what has previously been described in the literature, loss-to-follow-up and symptom-based antimuscarinic treatment could still lead to an overall underestimation of risk. However, sensitivity analyses showed that model performance was similar when persons who had antimuscarinics were also included. Moreover, loss-to-follow-up could lead to misleading results regarding the predictive power of the candidate predictors and suggests that the missing at random assumption underlying multiple imputations is unlikely to be fulfilled. However, the ‘informed’ sensitivity analyses with outcomes assigned based on further data from the clinical record, indicated that the modeling results are robust, in regard to loss-to-follow-up. Additionally, the study used the established cut-off for the risk of UUT deterioration of a storage pDetmax ≥ 40 cmH_2_O [6], although in the SCI population, evidence for this cut-off value is limited and contradictory, implying this cut-off might be too low [33] or too high [34]. Furthermore, no further specification of the type of neurogenic DO (phasic, terminal, compound, or sustained DO) or the quantification of DO (for example, measurement of the area under the curve or the DO index) [35] was utilized. However, these concepts harbor limitations and are, therefore, not implemented as standard in most neuro-urological units because current software to measure the area under the curve cannot exclude artifacts in urodynamic traces, while for the DO index, no standardized values are available [35]. A major limitation of this study was the small sample size, which precluded a thorough investigation of many of the targeted outcomes. Furthermore, it was not possible to include lag effects and change scores, despite the fact that clinical management is in the context of time-updated information. Finally, especially considering that a large proportion of the population already reached the outcomes at the first UDI timepoint, this study was not adequately powered to investigate the predictive value of the UDI findings from the 1-month timepoint.

## 5. Conclusions

Our study identified two clinical models that provided a fair, yet insufficient, prediction of the occurrence of high pDetmax (≥40 cmH_2_O) during the storage phase using data from routine neurological assessments of patients with SCI. We also showed that LEMS is not an adequate predictor of high storage pDetmax. Further prognostic model development research is warranted, as such models provide important support for the optimization of clinical practice, including patient consultations and planning. Models using sex, UEMS, SCI severity, and neurological level might provide adequate prognostic values for high storage pDetmax, to improve the design and analysis of clinical trials, provided that they can first be validated with an external dataset. Therefore, UDI remains an essential tool for the identification of patients with acute SCI and at risk of UUT deterioration due to high detrusor pressures during the storage phase.

## Figures and Tables

**Figure 1 biomedicines-11-01748-f001:**
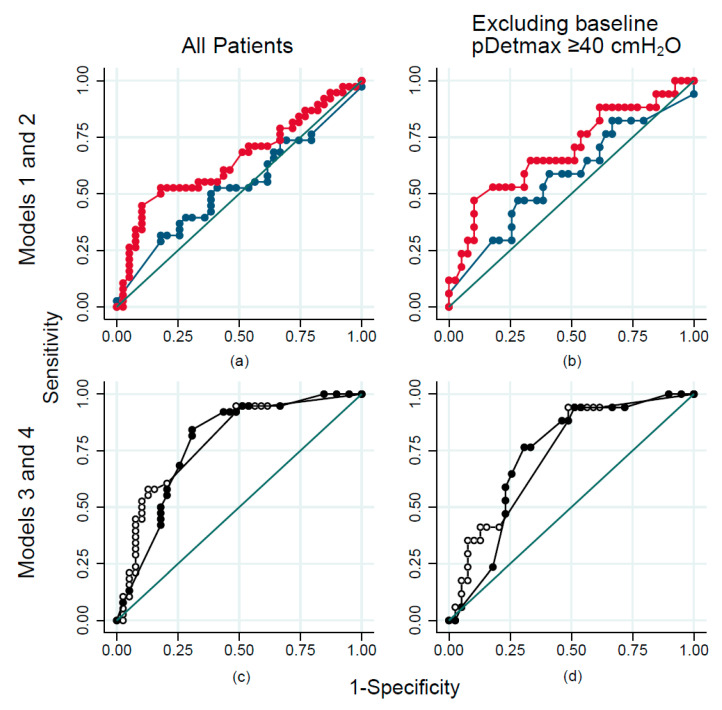
Area under the receiver operating curve (aROC) for prediction of maximum detrusor pressure (pDetmax) during the storage phase ≥ 40 cmH_2_O. Top: Models 1(blue) and 2 (red): Including all observations (**a**); omitting observations that had pDetmax ≥ 40 cmH_2_O at baseline (1 month after SCI) (**b**); bottom: Models 3 (white) and 4 (black): Including all observations (**c**); omitting observations that had pDetmax ≥ 40 cmH_2_O at baseline (**d**). Model 1: lower extremity motor score (LEMS); model 2: LEMS, highest light touch score of the S3 dermatome, Spinal Cord Independence Measure (SCIM) III respiratory–sphincter subscale; model 3: upper extremity motor score (UEMS) and sex; model 4: neurological level, American Spinal Injury Association Impairment Scale (AIS) grade and sex. The green line indicates the random classifier level, aROC=0.50.

**Figure 2 biomedicines-11-01748-f002:**
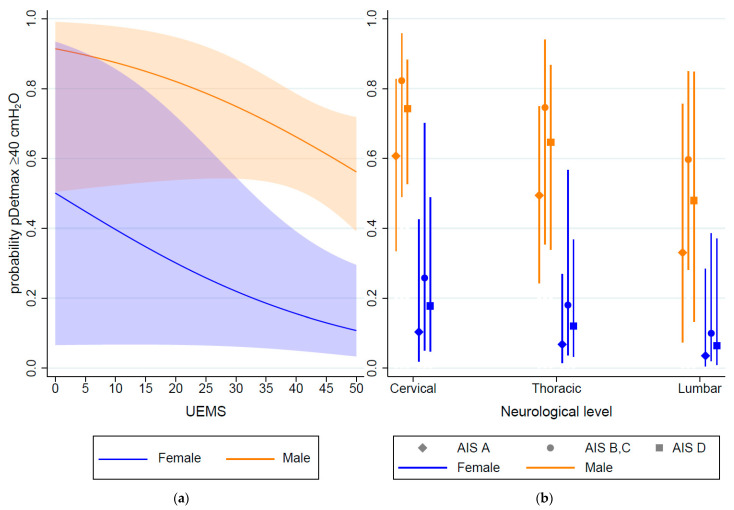
Predicted probability of maximum detrusor pressure (pDetmax) ≥ 40 cmH_2_O during the storage phase occurring in the first year after spinal cord injury (SCI), according to models 3 (**a**) and 4 (**b**). Both models showed fair discrimination (aROC 0.78–0.79). AIS: American Spinal Injury Association Impairment Scale (AIS) grade; UEMS: upper extremity motor score.

**Table 1 biomedicines-11-01748-t001:** Characteristics of the study population, overall, and stratified according to the availability of a 12-month urodynamic investigation (UDI) follow-up. Baseline variables reflect patient status within 40 days of a spinal cord injury (SCI) and outcome variables indicate the observation of the respective urodynamic parameter at any timepoint within the 1st year after a SCI.

Characteristic (% Missing)	Overall Population, N = 97	12-Month UDI Present, N = 73	Missed 12-Month UDI, N = 24	
**Baseline continuous variables**	**Median** **(Q1–Q3)**	**Median** **(Q1–Q3)**	**Median** **(Q1–Q3)**	***p*-value**
**Age at SCI (0)**	57 (42–69)	56 (45–69)	60 (40–72)	0.56
**Lower extremity motor score (LEMS) (5)**	33 (1–48)	32 (2–48)	40 (0–47)	0.98
**Upper extremity motor score (UEMS) (5)**	50 (38–50)	50 (41–50)	50 (27–50)	0.69
**SCIM Score (2)**	36 (22–68)	37 (22–74)	31 (22–59)	0.55
**SCIM Respiratory-Sphincter Subscale (2)**	15 (12–25)	15 (15–25)	14 (10–25)	0.32
**Baseline categorical variables**	**n (%)**	**n (%)**	**n (%)**	***p*-value**
**Sex (0)**				0.36
Female	33 (34)	23 (32)	10 (42)	
Male	64 (66)	50 (68)	14 (58)	
**Lesion etiology (0)**				0.17
Traumatic spinal cord injury	75 (77)	54 (74)	21 (87)	
Ischemic spinal cord injury	22 (23)	19 (26)	3 (13)	
**Neurological Level (5) ^a^**				0.88
Cervical (C1–C8)	43 (44)	33 (45)	10 (41)	
Thoracic (T1–T12)	37 (38)	28 (38)	9 (38)	
Lumbar (L1–L5)	17 (18)	12 (17)	5 (21)	
**SCI Severity (5) ^a^**				0.64
AIS A	21 (22)	18 (25)	3 (13)	
AIS B	10 (10)	7 (10)	3 (13)	
AIS C	16 (16)	12 (16)	4 (16)	
AIS D	50 (52)	36 (49)	14 (58)	
**S3 light touch score (6)**				0.45
Absent	25 (26)	21 (29)	4 (17)	
Altered	46 (47)	33 (45)	13 (54)	
Normal	20 (21)	14 (19)	6 (25)	
**Outcome** **variables**	**n (%)**	**n (%)**	**n (%)**	***p*-value**
**DO–DSD (0) ^b^**				<0.01
No	12 (12)	5 (7)	7 (29)	
Yes	85 (88)	68 (93)	17 (71)	
**pDetmax ≥ 40 cmH_2_O (0) ^b^**				<0.001
No	59 (61)	39 (53)	20 (83)	
Yes	38 (39)	34 (47)	4 (17)	
**Vesicoureteral reflux (1) ^b^**				0.12
No	89 (92)	66 (90)	23 (96)	
Yes	7 (7)	7 (10)	0 (0)	
**Any unfavorable UDI parameter (1) ^b^**				<0.01
No	9 (9)	3 (4)	6 (25)	
Yes	87 (90)	70 (96)	17 (71)	

^a^ When baseline data were missing, patients were assigned to a group based on information from the next testable timepoint; ^b^ indicating status at the time of loss-to-follow-up for individuals with missed 12-month UDIs. AIS: American Spinal Injury Association Impairment Scale grade: DO–DSD: detrusor overactivity with detrusor sphincter dyssynergia; pDetmax: maximum detrusor pressure (storage phase); S: sacral; SCIM: Spinal Cord Independence Measure Version III.

**Table 2 biomedicines-11-01748-t002:** Unfavorable urodynamic parameters stratified according to predictor characteristics. Bladder compliance < 20 mL/cmH_2_O did not occur among the patients in this dataset.

Characteristic (% Missing)	No DO–DSD	DO–DSD		No pDetmax ≥ 40 cmH_2_O	pDetmax ≥ 40 cmH_2_O		No VUR	VUR		No Unfavorable UDI Parameters	Unfavorable UDI Parameters	
**Outcome**	**n = 5**	**n = 85**		**n = 39**	**n = 38**		**n = 66**	**n = 7**		**n = 3**	**n = 87**	
**Continuous variables**	**Median** **(Q1–Q3)**	**Median** **(Q1–Q3)**	***p*-value**	**Median** **(Q1–Q3)**	**Median** **(Q1–Q3)**	***p*-value**	**Median** **(Q1–Q3)**	**Median** **(Q1–Q3)**	***p*-value**	**Median** **(Q1–Q3)**	**Median** **(Q1–Q3)**	***p*-value**
**Age at SCI (0)**	35 (34–67)	57 (47–71)	0.36	55 (36–71)	57 (45–66)	0.73	56 (49–70)	57 (32–69)	0.90	35 (34–67)	57 (45–71)	0.32
**LEMS (5)**	28 (0–50)	32 (2–47)	0.96	32 (5–48)	25 (0–50)	0.67	33 (3–50)	5 (0–16)	0.054	50 (0–50)	31 (2–47)	0.41
**UEMS (5)**	48 (15–50)	50 (36–50)	0.46	50 (47–50)	45 (21–50)	<0.01	50 (39–50)	50 (20–50)	0.92	50 (48–50)	50 (35–50)	0.43
**SCIM Respiratory-Sphincter subscale (2)**	62 (21–96)	34 (22–59)	0.42	38 (26–74)	35 (14–68)	0.26	38 (23–74)	27 (20–45)	0.45	92 (31–100)	34 (21–58)	0.091
**Categorical variables**	**n (%)**	**n (%)**	***p*-value**	**n (%)**	**n (%)**	***p*-value**	**n (%)**	**n (%)**	***p*-value**	**n (%)**	**n (%)**	***p*-value**
**Sex (0)**			0.65			<0.0001			0.42			>0.99
Female	1 (20)	31 (36)		20 (51)	3 (8)		22 (33)	1 (14)		1 (33)	31 (36)	
Male	4 (80)	54 (64)		19 (49)	35 (92)		44 (67)	6 (86)		2 (67)	56 (64)	
**Lesion etiology (0)**			>0.99			0.021			0.18			>0.99
Traumatic spinal cord injury	4 (80)	64 (75)		25 (64)	33 (87)		47 (71)	7 (100)		2 (67)	66 (76)	
Ischemic spinal cord injury	1 (20)	21 (25)		14 (36)	5 (13)		19 (29)	0 (0)		1 (33)	21 (24)	
**Neurological Level (5) ^a^**			0.38			0.23			0.51			>0.99
Cervical (C1–C8)	4 (80)	36 (42)		14 (36)	21 (55)		31 (47)	2 (29)		2 (67)	38 (44)	
Thoracic (T1–T12)	1 (20)	34 (40)		17 (44)	12 (32)		25 (38)	3 (42)		1 (33)	34 (39)	
Lumbar (L1–L5)	0 (0)	15 (18)		8 (20)	5 (13)		10 (15)	2 (29)		0 (0)	15 (17)	
**SCI Severity (5) ^a^**			0.10			0.40			0.10			0.61
AIS A	3 (60)	18 (21)		10 (26)	8 (21)		15 (23)	3 (43)		1 (33)	20 (23)	
AIS B/C	0 (0)	23 (27)		8 (20)	13 (34)		16 (24)	3 (43)		0 (0)	23 (26)	
AIS D	2 (40)	44 (52)		21 (54)	17 (45)		35 (53)	1 (14)		2 (67)	44 (51)	
**S3 light touch score (6)**			0.27			0.60			0.13			0.065
Absent	2 (40)	23 (27)		11 (28)	10 (26)		17 (26)	1 (14)		1 (33)	24 (28)	
Altered	1 (20)	42 (49)		20 (51)	16 (42)		32 (48)	2 (29)		0 (0)	43 (49)	
Normal	2 (40)	15 (18)		6 (15)	9 (24)		12 (18)	0 (0)		2 (67)	15 (17)	

^a^ When baseline data were missing, patients were assigned to a group based on information from the next testable timepoint; AIS: American Spinal Injury Association Impairment Scale grade: DO–DSD: detrusor overactivity with detrusor sphincter dyssynergia; LEMS: lower extremity motor score; pDetmax: maximum detrusor pressure (storage phase); S: sacral; SCI: spinal cord injury; SCIM: Spinal Cord Independence Measure Version III; UDI: urodynamic investigation; UEMS: upper extremity motor score; VUR: vesicoureteral reflux.

## Data Availability

The data presented in this article are available on request from the corresponding author.

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
