# Peer review of "Urodynamics Are Essential to Predict the Risk for Upper Urinary Tract Damage after Acute Spinal Cord Injury"

_biomedicines, 2023, doi:10.3390/biomedicines11061748_

Round 1
Reviewer 1 Report
This manuscript reported using two clinical models might provide a fair prediction of the occurrence of high Pdetmax (≥40 cmH2O) during the storage phase using data from routine neurological assessments of patients with SCI. Models using sex, UEMS, SCI severity, and neurological level might provide adequate prognostic value for high storage Pdetmax to improve the design and analysis of clinical trials. However, the authors concluded that UDI remains essential and cannot be replaced for the identification of patients with acute SCI at risk for upper urinary tract deterioration due to high detrusor pressures during the storage phase.
This study is interesting, but the results are insufficient to draw the conclusion. There are several factors which might influence the development of high PdetQmax, such as poor bladder management, high level SCI with inhibition of detrusor contraction, prolonged spinal shock stage, and multiple SCI injury levels. Although the AUC for ROC is satisfactory (>0.7), the results might not be applied to clinical reference.
Currently, a urodynamic study, if possible a videourodynamic study, remains an essential investigating tool to assess the urinary tract function of SCI patients and may guide the management strategy to prevention of upper urinary tract deterioration.
Reviewer 2 Report
The manuscript is well-written and presented. Two points the authors need to address:
- The significance of the maximum detrusor storage pressure as a predictor of upper urinary tract damage (evidence, how it is correlated), as the study was based on this parameter as an outcome.
- Add a comment if these patients were assessed for any signs of upper urinary tract symptoms (these symptoms need specification as well) or if the analyses relied completely on the assumption that pDetMax was the primary outcome.
Thank you
